# Competitive binding and molecular crowding regulate the cytoplasmic interactome of non-viral polymeric gene delivery vectors

Aji Alex M. Raynold[1,2,3], Danyang Li[1,2,3], Lan Chang [1,2] & Julien E. Gautrot [1,2✉]

In contrast to the processes controlling the complexation, targeting and uptake of poly-cationic gene delivery vectors, the molecular mechanisms regulating their cytoplasmic dissociation remains poorly understood. Upon cytosolic entry, vectors become exposed to a complex, concentrated mixture of molecules and biomacromolecules. In this report, we characterise the cytoplasmic interactome associated with polycationic vectors based on poly(dimethylaminoethyl methacrylate) (PDMAEMA) and poly(2-methacrylolyloxyethyl-trimethylammonium chloride) (PMETAC) brushes. To quantify the contribution of different classes of low molar mass molecules and biomacromolecules to RNA release, we develop a kinetics model based on competitive binding. Our results identify the importance of competition from highly charged biomacromolecules, such as cytosolic RNA, as a primary regulator of RNA release. Importantly, our data indicate the presence of ribosome associated proteins, proteins associated with translation and transcription factors that may underly a broader impact of polycationic vectors on translation. In addition, we bring evidence that molecular crowding modulates competitive binding and demonstrate how the modulation of such interactions, for example via quaternisation or the design of charge-shifting moieties, impacts on the long-term transfection efficiency of polycationic vectors. Understanding the mechanism regulating cytosolic dissociation will enable the improved design of cationic vectors for long term gene release and therapeutic efficacy.

[1] Institute of Bioengineering, Queen Mary, University of London, Mile End Road, London E1 4NS, UK. [2] School of Engineering and Materials Science, Queen Mary, University of London, Mile End Road, London E1 4NS, UK. [3] These authors contributed equally: Aji Alex M. Raynold, Danyang Li. ✉email: j.gautrot@qmul.ac.uk

The delivery of RNA therapeutics has rapidly progressed in the last two decades, enabling the development of gene silencing, microRNA delivery for tissue engineering and cancer therapies and the development of RNA-based vaccines[1–3]. Amongst the range of viral and non-viral vectors that have been designed, polycationic systems are particularly attractive for in vitro work[4–7], although their translation and efficacy in vivo has been limited. To some extent, this is due to the more difficult control of biodistribution and oligonucleotide delivery that can be achieved with these systems, in comparison with lipid-based and viral systems[8]. Upon cellular uptake, rapid release from polymeric vectors can be modulated by a variety of stimuli and processes, including light, exposure to glutathione and hydrolytic degradation of charge shifting systems, typically aiming to speed up release[9,10]. However, sustained release from polycationic vectors remains an important process to control as it will enable longer-term therapeutic effects and will limit repeated injections and exposure to high burst doses. Similarly, in vitro sustained release will constitute an attractive alternative to knockdown via a shRNA strategy whilst retaining stable silencing or microRNA signalling.

Understanding the release mechanism of RNA from poly-cationic vectors appears key to achieve long term silencing, signalling and ultimately therapeutic efficacy. Although the strength of complexation, the structure of the resulting complexes and their uptake and cytosolic entry have been widely studied[11–14], our understanding of the release of oligonucleotides from polycation complexes remains limited and qualitative. Plasmid DNA was found to release rapidly, within seconds, upon cytosolic entry, confirming the short burst effect typically achieved[15]. This may be appropriate for DNA delivery requiring fast release to enable nuclear translocation and expression, however, such behaviour severely limits the sustained release of RNA. In particular, the molecular processes regulating complex dissociation in the cytoplasm and the composition of the associated cytosolic interactome remain unexplored.

In this report, we characterise the cytoplasmic proteome associated with polycationic gene delivery vectors (poly (dimethylaminoethylmethacrylate) (PDMAEMA) and poly(2-methacrylolyloxyethyltrimethylammonium chloride) (PMETAC) brush-functionalised nanoparticles). We propose that competitive binding is underlying the formation of the associated interactome, regulated by molecular crowding, and develop a kinetics model describing the displacement of RNA molecules by competitors. Finally, we demonstrate that engineering of the binding strength of oligonucleotides enables the modulation of competitive binding and the sustained release and knockdown efficiency of siRNA.

## Results

We focused on PDMAEMA and PMETAC brush-functionalised nanoparticles as cationic model systems. PDMAEMA-based vectors have been widely studied and applied for gene delivery[16,17]. The high density of polymer brushes was found to bind RNA oligonucleotides particularly stably[18], yet display low cytotoxicity and be amenable to molecular design, including labelling of corresponding nanomaterials[19]. The high density of polymer brushes was found to impact not only the binding capacity of these coatings, but also their kinetics of adsorption, depending on the molecular weight of the macromolecules adsorbed[20]. Hence, this system allows the tailoring of physico-chemical parameters regulating RNA adsorption and transfection.

To determine which cytoplasmic proteins interact with cationic polymers, we allowed polymer brush-functionalised nanoparticles to interact with cytosolic fractions, prior to their separation via centrifugation and washing. Following desorption of the adsorbed molecules and digestion, protein analysis via mass spectrometry was performed and compared to the composition of pristine cytosolic fractions (Fig. 1a, b). Proteomic analysis of the pristine cytosolic fractions confirmed the presence of expected cytoplasmic proteins, including a large pool of proteins associated with cytoskeleton assembly and organisation, either originating from monomeric soluble proteins, or potentially oligomers that may not have been separated in our fractionation protocol (in total, 376 proteins, with half of the functional clusters identified being associated with the regulation of cytoskeleton assembly; Supplementary Table 1). In contrast, we identified 89 proteins present in the PDMAEMA/PMETAC brush-cytosolic proteome. 77 of these proteins are predicted to display a low isoelectric point (pI, predicted average of 5.25), below 7.0 and the pKa of PDMAEMA brushes, therefore potentially adsorbing to PDMAEMA and PMETAC brushes via electrostatic interactions (Fig. 1b). However, 12 of the proteins identified in the vector proteome (13% of the total) were predicted to display relatively high isoelectric points (as high as 11.6, Fig. 1b).

Although the adsorption of proteins to like-charged polymer brushes has been reported[21], we suspected that high pI proteins adsorbed indirectly to polycationic brushes. Analysis of the function of the highly abundant proteins identified in the vector interactome led us to group them into 4 main categories (Fig. 1a). The first category consisted in proteins involved in translation or directly binding RNA or DNA, including the neuroblast differentiation-associated protein (AHNK), heterogeneous nuclear ribonucleoprotein K (HNRPK) and eukaryotic translation initiation factor 4H (IF4H) and several histones (with high pIs). Two other proteins identified with particularly high abundance and displaying high basicity were activated RNA polymerase II transcriptional coactivator p15 (TCP4; predicted pI of 10.0) and 40S ribosomal protein S21 (RS21; predicted pI of 8.3), associated with the regulation of transcriptional[22] and translational activity[23]. Most of these proteins displayed relatively low abundance in the corresponding cytosolic fractions or lysates and were clearly enriched in the PDMAEMA/PMETAC brush-cytoplasmic proteome. Another group of proteins identified, although typically with lower abundance, was associated with endosomes, the proteasome and lysosomes. These included caveolae-associated protein 1 and clathrin light chains A and B, associated with endocytosis[24], and E3 ubiquitin-protein ligase PPP1R11, involved in the regulation of proteolytic degradation[25], or reticulocalbin-1, involved in the regulation of the endoplasmic reticulum functioning[26]. A third group of proteins, particularly abundant, are cytoskeleton associated proteins, typically relatively acidic proteins with low predicted isoelectric points, and abundant in the cytosol (identified with high QE in cytosolic fractions). Therefore, these proteins may associate via electrostatic interactions and are detected due to their high background level in the cytosol. However, polymer brush-based cationic vectors may also associate to the cytoskeleton via the endosome or microtubules (such as microtubule associated proteins 1A, 1B and 4 or kinesin) during transport towards the endoplasmic reticulum and nucleus. A final group of proteins was generally associated with the cytoplasm, reflecting molecular composition in the cytosolic fractions isolated.

The distribution observed within the most abundant pool of proteins was also well reflected by cluster analysis (Supplementary Fig. 1), with many of the proteins identified in the first two groups clustering together, and highly enriched in the polycation proteome compared to the pristine lysate. Further functional clustering analysis identified 6 gene groups (Supplementary Table 2) associated with the first three main groups of proteins identified in Fig. 1a. In particular, the two groups with highest enrichment scores were associated with RNA/DNA binding,

ARTICLE

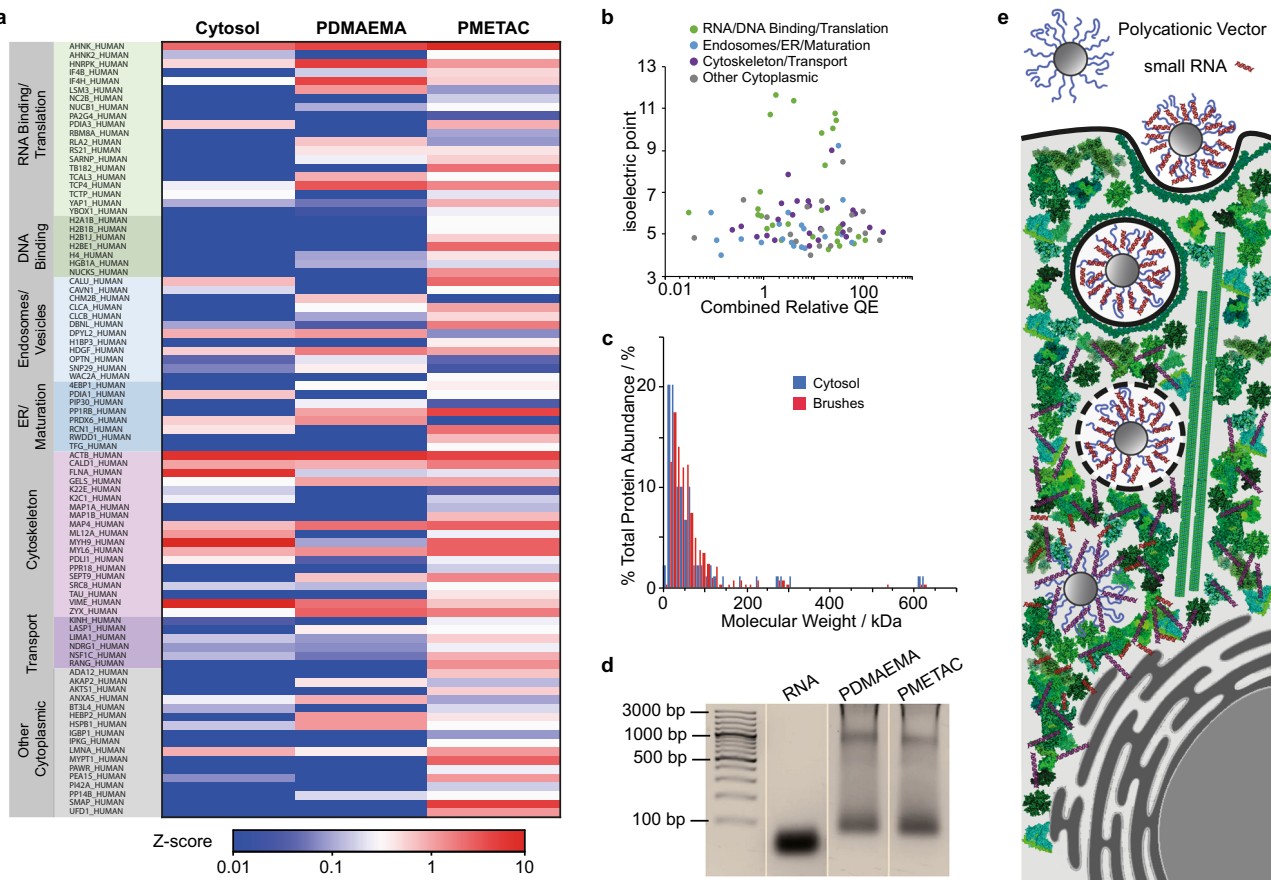

**Fig. 1 Analysis of the polycationic vector interactome. a** Most abundant proteins identified by proteomics analysis of cytosolic adsorbates to PDMAEMA- and PMETAC-brush grafted nanoparticles, compared to protein abundance in original cytosolic fractions. **b** Correlation between the isoelectric point of the corresponding proteins and their abundance in the polycationic vector proteome. **c** Molecular weight distribution of proteins identified by proteomic analysis of cytosolic fractions (blue) and brush-adsorbate (red). **d** Agarose gel electrophoresis analysis of nucleic acids bound to polymer brushes (PDMAEMA and PMETAC; representative from triplicate experiments) after incubation in cell lysates (bound nucleic acids were desorbed via incubation in 2 M NaCl). The siRNA sample (22 bp) used for initial adsorption is shown for comparison. **e** Competitive binding in a crowded molecular environment is proposed to regulate RNA release upon cytoplasmic entry.

translation, the endosome, endoplasmic reticulum and post-translational modification. In contrast, functional clustering on the proteome identified in cytosolic fractions revealed that the 5 groups with highest enrichment were associated with the cytoskeleton and cargo transport. We also noted that comparable analysis of the proteome associated with PDMAEMA brush gene delivery vectors upon interaction with unfractionated cell lysates, although less relevant as containing a broader range of macromolecules and organelles, resulted in comparable functional analysis and the identification of proteins displaying high isoelectric points (see Supplementary Discussion 1, Supplementary Figs. 2 and 3 and Supplementary Table 3). Therefore, the proteome associated with polycationic gene delivery vectors appears enriched in proteins involved in RNA binding, translational activity, post-translational modification, endoplasmic reticulum localisation and vesicle (endosomes, lysosome) localisation and transport. We note that no significant shift in the molecular weight distribution of the polycationic vector proteome was observed compared to that of cytosolic fractions (Fig. 1c).

The function of the largest pool of proteins identified with the PDMAEMA/PMETAC brush vector cytoplasmic proteome, associated with translation and RNA/DNA binding, and the high isoelectric point of some of these proteins suggested that these macromolecules may associate indirectly with polycationic brush vectors, potentially via cytosolic RNA. Indeed, gel electrophoresis

analysis clearly indicated the occurrence of oligonucleotides bound to vectors from lysates, with a molecular weight near 100 bp (Fig. 1d). A higher band also occurs near 1000 bp and a large smear of moderate intensity bridged between these two main bands. Hence, a wide distribution of polynucleotides with molecular weights significantly higher than that of the small oligonucleotides typically used for siRNA delivery was found to be associated with the polycation cytoplasmic interactome. In addition, although our data does not provide direct evidence that part of the proteome identified is recruited via RNA binding, we observed that basic macromolecules such as poly(L-lysine) (tagged with Alexa Fluor™ 594), as well as acidic macromolecules such as BSA (tagged with Alexa Fluor™ 594) can indeed adsorb to PDMAEMA brush-functionalised nanoparticles loaded with RNA (tagged with 6-FAM; Supplementary Fig. 4). This phenomenon was associated with a marked displacement of the RNA oligonucleotides initially loaded into the vectors (although not complete), suggesting that competitive interactions lead not only to maturation of the biochemical composition of the vector corona, but also result in the desorption of RNA molecules.

Our analysis of the polycation cytoplasmic interactome, therefore, suggested that competitive binding with cytoplasmic molecules and macromolecules, in particular, may regulate the cytosolic release of oligonucleotides for gene delivery (Fig. 1e). To quantify the release rate from PDMAEMA brushes upon

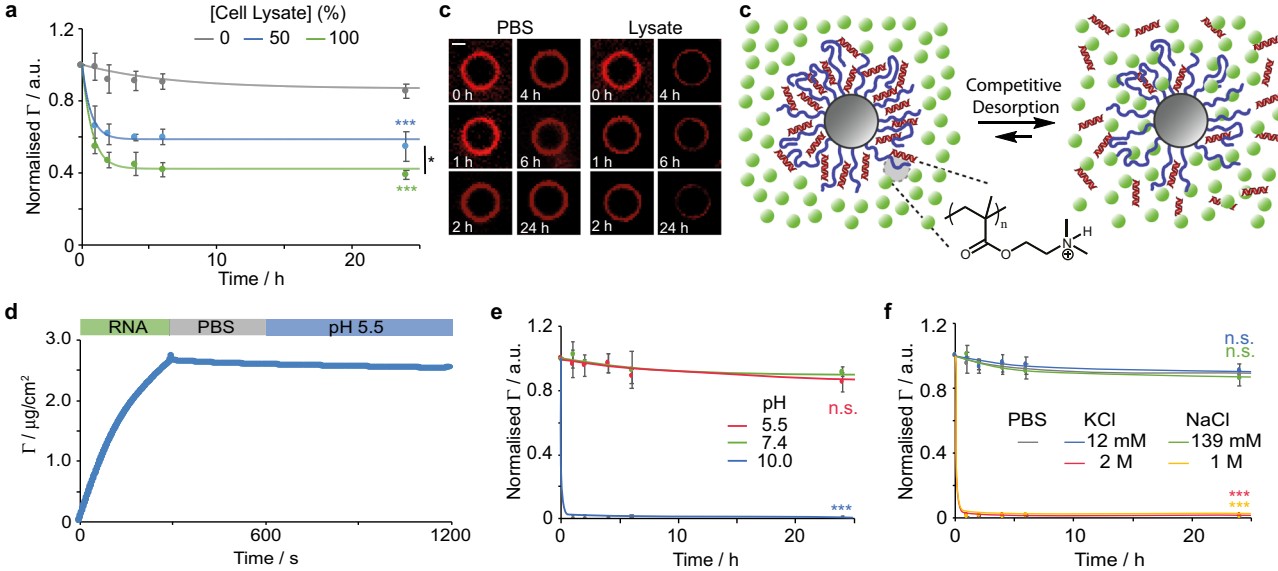

**Fig. 2 Rapid desorption of RNA in cell lysates. a** Release profile of cy-5 tagged siRNA from PDMAEMA brush-functionalised microparticles in cell lysate-PBS mixtures (v/v %). **b** Corresponding fluorescence microscopy images (scale bar, 1 μm). **c** Schematic illustration of competitive desorption of siRNA from cationic PDMAEMA brushes. **d** Quantification of siRNA adsorption (10 μg/mL; 50 μL injected at 10 μL/min flow rate) on PDMAEMA brushes, measured by SPR, followed by washing in PBS at neutral pH and pH 5.5. Release profiles of cy5 siRNA from PDMAEMA brush-functionalised microparticles at different pH (in PBS, **e**) and electrolytes (**f**). a.u., arbitrary units. Data are presented as mean values. Error bars are standard errors from triplicate experiments. n.s., not significant; *$p < 0.05$; ***$p < 0.001$ (ANOVA); with respect to 0% (**a**), pH 7.4 (**e**) and PBS (**f**), unless indicated otherwise.

cytoplasmic entry, we used microparticles and monitored the release of Cy5-tagged RNA (22 bp) in cell lysates, by microscopy (Fig. 2a–c). A maximum surface density of RNA adsorbed within 30 nm brushes of 2.6 μg/cm$^2$ was measured by SPR (Fig. 2d), which had previously been found to be dependent on thickness and grafting density[20]. Whereas the fluorescence intensity around particles only modestly decreased in PBS over the course of 24 h, we observed the desorption of over 60% of the total amount of RNA in undiluted cell lysate, and an intermediate level in 50% diluted lysate.

We confirmed that the release of RNA was not triggered at low pH (5.5), via SPR and our fluorescence assay, but rapidly occurred in basic pH (Fig. 2e). Stability was also observed at physiological ionic strength but not at ionic strength of 1 and 2 M, for KCl and NaCl electrolytes, respectively (Fig. 2f). Hence, under cytosolic and endosomal physiological conditions, RNA binding is stable and we proposed that desorption in lysates is regulated by competition with molecules and macromolecules present in such crowded environments (Fig. 2c).

In order to study such competitive desorption, we developed a kinetic model, extending from a validated adsorption model[20] capturing the response of oligonucleotide adsorption to the architecture of polymer brushes (Fig. 3a). In this model, we introduce a competitive equilibrium for the exchange of RNA with surrounding molecules at the surface of the brush and an equilibrium regulating the diffusion of the competing molecules. This leads to an exponential decrease of the surface density of bound oligonucleotides Γ as a function of time (see Supplementary Discussion):

$$\Gamma = \frac{k_{app1} C_S [Br]_0 - k_A [A_{bulk}]}{k_{app3}} - \left( \frac{k_{app1} C_S [Br]_0 - k_A [A_{bulk}]}{k_{app3}} - \Gamma_{max} \right) e^{-k_{app3}}$$

(1)

We first investigated the role of small molecule competitors. Exchange with positively charged amino acids, which may compete for binding with oligonucleotides, had negligible effects on desorption at physiologically relevant concentrations

(30–40 mM)[27] (Fig. 3b). At high concentrations, arginine and lysine (1 and 2 M, respectively) led to fast desorption to levels comparable to those of high ionic strength solutions, suggesting that electrostatic screening is the key element via which single amino acids may contribute.

The induction of desorption by short peptides such as the tripeptide glutathione, present at relatively high concentrations in the cytoplasm (1–10 mM)[28], was investigated next. At physiological concentrations (10 mM), competitive desorption was modest, although its contribution increased at higher concentrations, with a relatively slow desorption kinetics (Fig. 3c). Hence we conclude that small negatively charged peptides, and potentially other low molar mass molecules, can contribute to a modest level to RNA exchange, although these are unlikely to account for the level and kinetics of desorption observed for lysates.

Competition with macromolecules was investigated next. In agreement with the effective displacement typically reported with heparin[12,13], a fast desorption was observed even at concentrations as low as 5 μg/mL and was nearly complete above 25 μg/mL (Fig. 3d). Similarly, hyaluronic acid (HA) and chondroitin sulfate (CS) induced RNA desorption, although at slightly higher concentrations (above 100 and 25 μg/mL, respectively; see Supplementary Fig. 5). In comparison, when complexes were incubated in bovine serum albumin (BSA) solutions, moderate displacement was only observed, even at concentrations above 40 mg/mL (Fig. 3e). Hence strong polyelectrolytes with high charge densities are found to induce faster desorption and displacement of oligonucleotides from PDMAEMA brushes. Glycosaminoglycans are expressed at relatively high concentrations in the cytoplasm (in keratinocytes, 10 mg/mL levels of HA were reported[29]). Although this correspond to proteoglycans as well as free glycosaminoglycan chains, analysis of their respective distribution indicates comparable abundance and molecular weights in the range of 10 kDa[30]. Sulfated glycosaminoglycans were found to be particularly abundant in keratinocytes, with chondroitin sulfate expressed at 5-fold levels compared to those of HA[31]. Therefore, we estimate the cytoplasmic glycosaminoglycan

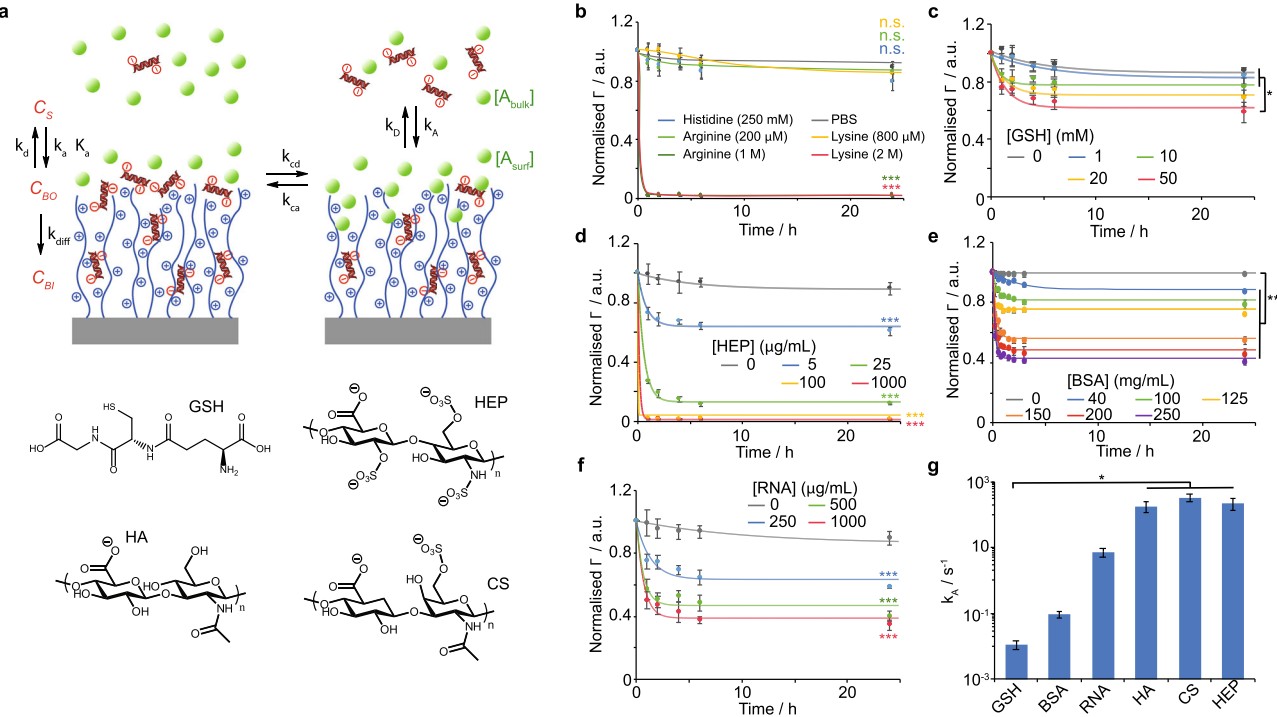

**Fig. 3 Competitive binding regulates RNA exchange in physiological conditions. a** Model of competitive binding with siRNA infiltrated within polymer brushes. **b–f** Release profiles of cy-5 tagged siRNA from PDMAEMA brush grafted silica microparticles in the presence of various amino acids (**b**; lines are only intended as guide for the eye), glutathione (**c**), heparin (**d**), BSA (**e**) and RNA (**f**), as a function of competitor concentration. Lines shown in (**c–f**) are fit lines based on Eq. 1. **g** Measured adsorption rate constant ($k_A$) of different competitors extracted from Supplementary Equation 11. a.u., arbitrary units. Data are presented as mean values. Error bars are standard errors from triplicate experiments. n.s., not significant; *$p < 0.05$; **$p < 0.01$; ***$p < 0.001$ (ANOVA); with respect to PBS (**b**) and 0 mg/mL (**d–f**), unless indicated otherwise.

concentration to be in the range of 0.1–1 mg/mL. The contribution of intracellular glycosaminoglycans to cytoplasmic release can therefore be expected to be very significant. In comparison, bovine serum albumin, despite its low isoelectric point (4.7), led to more modest, although non-negligible, displacement at a concentration comparable to that of protein cytosolic concentrations[32].

We next investigated the impact of cytosolic RNA on the dissociation of complexes. To mimic the cytosolic concentration of RNA (in the mg/mL range[33]), we exposed oligonucleotide-loaded PDMAEMA vectors to concentrations of 22 bp RNA in the range of 250–1000 μg/mL. We observed a rapid desorption of oligonucleotides from PDMAEMA-brush complexes, dependent on concentrations and leading to 60% release in 1 mg/mL RNA solutions (Fig. 3f). Analysis of the kinetics data obtained for the different molecular and macromolecular competitors (from Eq. 1 and Supplementary Equation 11) indicated an intermediate rate of adsorption for RNA competitors, compared with albumin and hyaluronic acid, but significantly lower than that measured for heparin (Fig. 3g). Hence displacement is clearly regulated by the strength and charge density of the macromolecules competing, as well as their molecular weight.

Finally, we investigated whether molecular crowding within the cytosol may impact on competitive binding and modulate oligonucleotide desorption. RNA-loaded PDMAEMA-brush microparticles were incubated in solutions of heparin, at a concentration at which rapid release was observed (20 μg/mL, Fig. 3d), containing increasing concentrations of BSA (molecular crowding agent, taken as a typical cytosolic protein with moderately acidic pI, in the range of 40–250 mg/mL). Although the ultimate level of oligonucleotide release (and $\Gamma_\infty$) was unaffected by molecular crowding (Supplementary Fig. 6A), the rate at

which desorption was observed was found to decrease as a function of increasing crowding (Supplementary Fig. 6B). Hence the apparent constant of release $k_{app3}$ gradually increased when the concentration of BSA increased in the range of 40 to 250 mg/mL, a range typically associated with cytoplasmic protein concentrations[34]. These observations imply that molecular crowding within the cytosol, and potentially in other micro-environments experienced by vectors during gene delivery, has an impact on the release of genetic materials to be delivered. However, as with other examples of molecular crowding[32,35], it is not clear whether this effect arises from limited diffusion or enhanced affinity of RNA for brushes.

Overall, our data indicate that a range of molecules and macromolecules significantly contribute to oligonucleotide displacement from non-viral polycationic vectors (Fig. 1e). Although individual contribution from small molecules is weak, their global contribution may be expected to be non-negligible. Similarly, proteins with low pI contribute to the displacement of oligonucleotides, although weakly. RNA macromolecules, although intermediate competitors compared to densely charged glycosaminoglycans, present at high concentrations in the cytosol, are able to effectively displace oligonucleotides to be delivered and contribute to the formation of a broader interactome that involves translation complex proteins and RNA/DNA binding proteins. The direct implication of such competitive displacement in a complex molecular environment is that the engineering of vectors stabilising oligonucleotides and tuning their competitive desorption should allow the sustained delivery and prolonged therapeutic efficacy of small RNA.

To test this hypothesis, we modified PDMAEMA brushes to modulate the strength of their interaction with RNA and their release (Fig. 4a). Charge shifting brushes (CS-PMETAC) were

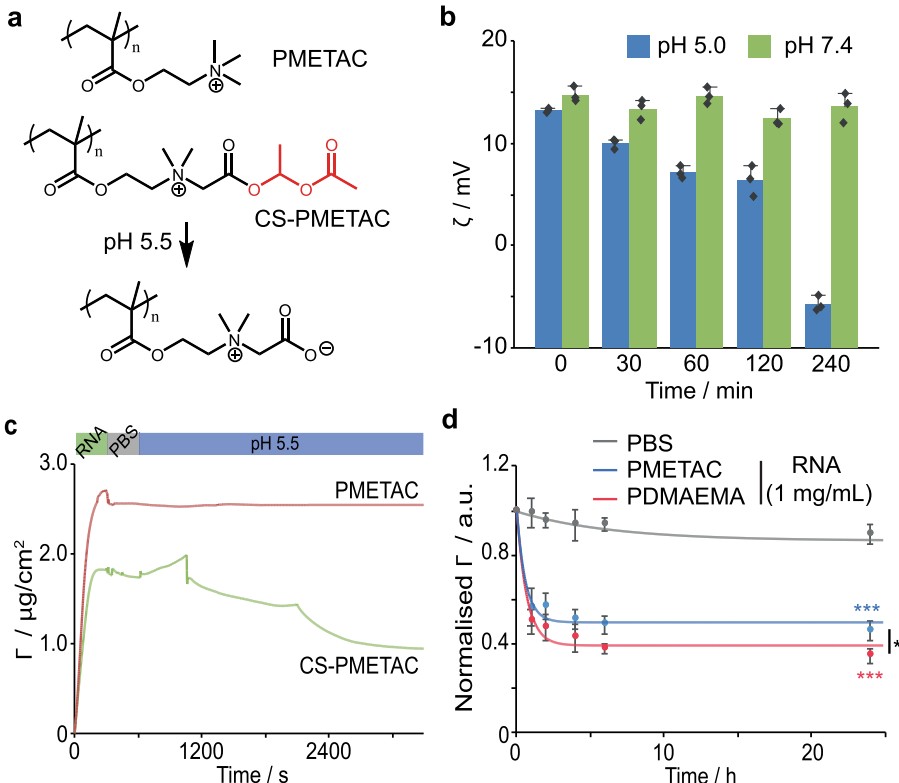

**Fig. 4 Design of engineered polymer brush vectors regulating competitive binding. a** Structure of PMETAC and CS-PMETAC brushes. **b** Change in $\zeta$-potential of silica nanoparticles coated with CS-PMETAC upon incubation in pH 5.0 and pH 7.4 PBS for different time points, up to 4 h. **c** Quantification of siRNA adsorption (10 µg/mL; 50 µL injected at 10 µL/min flow rate) on PMETAC and CS-PMETAC brushes, measured by SPR, followed by washing in PBS at neutral pH and pH 5.5. **d** Release profile of cy5 tagged siRNA from PDMAEMA and PMETAC brush-grafted microparticles in the presence of competing RNA (1 mg/mL in PBS). a.u., arbitrary units. Data are presented as mean values. Error bars are standard errors from triplicate experiments. n.s., not significant; *$p < 0.05$; ***$p < 0.001$ (ANOVA); with respect to PBS (**d**), unless indicated otherwise.

generated by quaternisation with hydrolysable 1-acetoxyethyl-2-bromoacetate residues (Supplementary Figs. 7–14). This afforded CS-PMETAC brushes that degraded within 4 h, at endosomal pH (5.0-5.5), and led to the charge inversion of corresponding vectors (Fig. 4b, c). In contrast, quaternisation with methyl iodide led to the formation of strong polyelectrolyte PMETAC brushes, which bound comparable levels of RNA to PDMAEMA and retained their load even at endosomal pH. Competitive assay with RNA revealed that these brushes led to slower release kinetics compared to PDMAEMA (Fig. 4d).

To investigate the fate of delivered RNA upon cytosolic entry, we transfected HaCaT cells with tagged oligonucleotides (Fig. 5 and Supplementary Figs. 15–17). To visualise the localisation of vectors, we introduced a tag in the initiator layer, therefore enabling to retain the surface physico-chemistry of polymer brush-based vectors[19]. We observed that CS-PMETAC vectors rapidly released oligonucleotides and little co-localisation of RNA and vectors could be observed compared to PDMAEMA particles (Fig. 5a–f). In contrast, RNA remained more strongly associated with PMETAC brushes, in agreement with the restricted competition with cytoplasmic macromolecules. This was despite the comparable level of endosomal escape of PMETAC and PDMAEMA particles.

In turn, we investigated the impact of quaternisation on knockdown efficiency. We used a model of GFP-actin expressing HaCaT cells and compared the expression of GFP to that of endogenous actin. Cells were subjected to a single transfection at day 0. In agreement with our previous report, PDMAEMA-based vectors led to comparable levels of knockdown to lipofectamine at

early time points (Fig. 5g and Supplementary Fig 17). The performance of PDMAEMA-brush based vectors indicates that, the surface availability of RNA, which could be expected to lead to rapid competitive desorption, is restricted by the entanglement associated with high brush density[36]. In comparison, CS-PMETAC resulted in relatively low knockdown efficiency, consistently with their reduced RNA adsorption and rapid dissociation (Fig. 5g and Supplementary Fig. 17). The knockdown efficiency remained high in the case of PDMAEMA for 4 days, and 3 days for lipofectamine (near 66%), prior to reduction to 52% at day 4. In contrast, PMETAC brushes led to an initial weaker knockdown (after 24 h, 42 and 52% for N/P ratios of 10 and 20, respectively), but this increased to 70 and 79% efficiencies (for N/P ratios of 10 and 20, respectively), which was sustained until 4 days of culture. To explore the longer term knockdown efficiency of PMETAC brushes, we compared lipofectamine and PMETAC brush-based transfection 10 days after transfection (Fig. 5h–i). Knockdown levels remained high (66.9 ± 3.7%) with PMETAC brushes, even after such long culture period. In comparison, lipofectamine knockdown levels remained low (51%). The retention of a moderate knockdown efficacy at such time point with lipofectamine may indicate that a pool of proteins (in our model, actin) is not recycled as fast and may persist, resulting in a global decrease in non-tagged proteins, or could also reflect the reduction in cell cycling that is typically observed as cells reached confluency (although cells were passaged at day 5 and cycling continued afterwards). Hence the modulation of RNA displacement in polymer brush-based vectors enables the sustained delivery of siRNA and led to prolonged knockdown in our model.

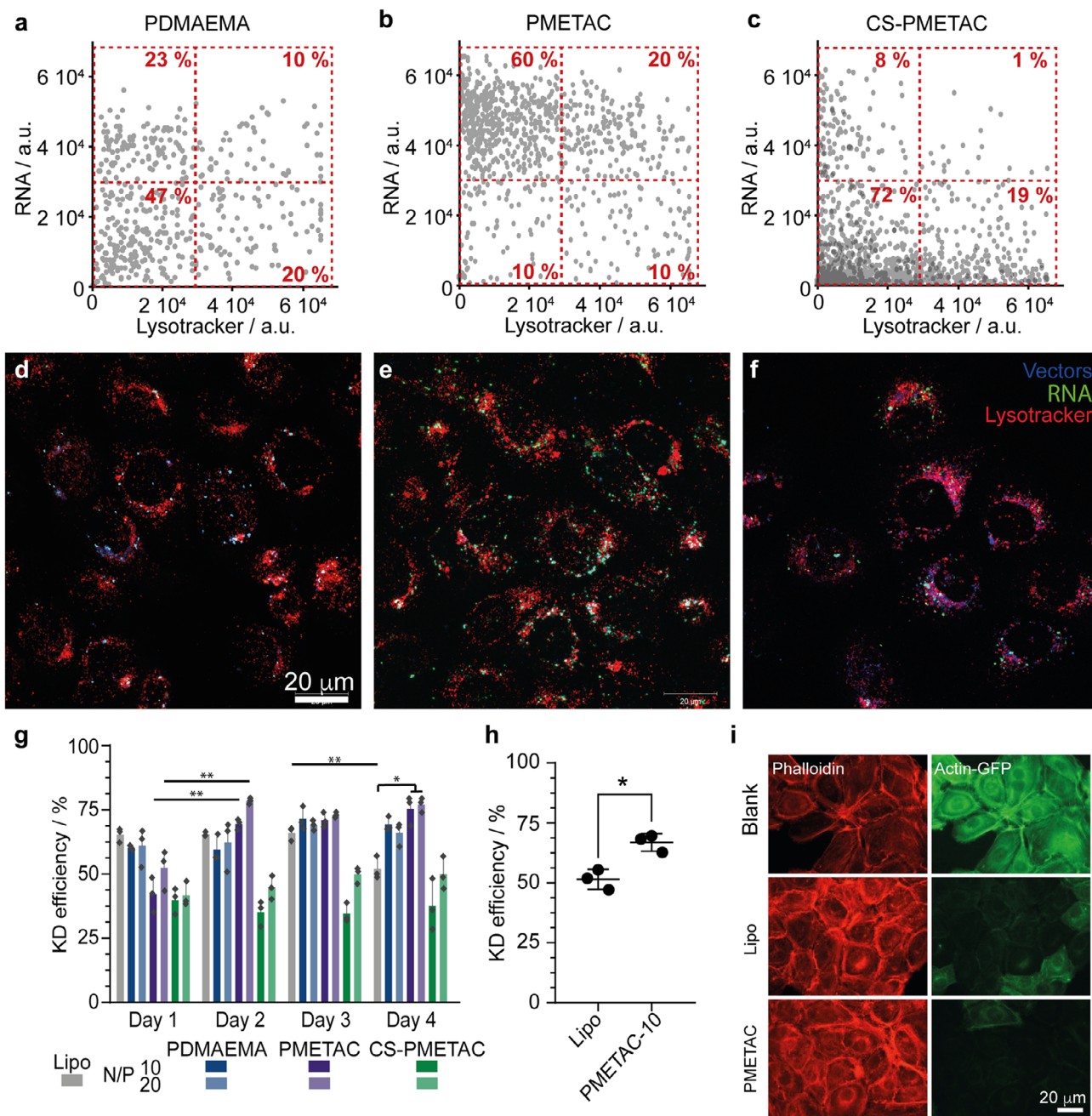

**Fig. 5 Brushes with higher RNA retention allow long term transfection with high efficiency. a–c** Co-localisation of siRNA and lysosomes/endosomes (lysotracker) in cells transfected with PDMAEMA, PMETAC and CS-PMETAC (HaCaT cells transfected for 2 days). Numbers in dashed boxes correspond to the quantification of corresponding populations. **d–f** Corresponding fluorescence microscopy images (blue, nanoparticles; green, siRNA; red, lysotracker). **g** HaCaT actin-GFP cell transfection efficiency (GFP siRNA) with PDMEAMA, PMETAC and CS-PMETAC functionalised nanoparticles (N/P ratios of 10 and 20), compared to lipofectamine 2000 (Lipo). Time points 1–4 days. **h** Long-term (day 10) transfection efficiency of PMETAC-functionalised nanoparticles and lipofectamine. **i** Representative images of untreated HaCaT actin-GFP control and cells 10 days after transfection with lipo and PMETAC-functionalised nanoparticles. Results of three independent experiments. a.u., arbitrary units. Data are presented as mean values. Error bars are standard errors from triplicate experiments. n.s., not significant; *$p < 0.05$; **$p < 0.01$ (ANOVA); with respect to 0 % (**a**), pH 7.4 (**e**) and PBS (**f**), unless indicated otherwise.

## Discussion

The model presented captures the competitive binding of polymer brush-RNA complexes with free solutes with a range of molecular structures and molecular weights. The extent of such competition correlates with the strength of the competing polyelectrolyte and its charge density, but also molecular weight. This is in agreement with electrostatic adsorption at the brush surface

playing a dominating role in competitive binding. We observed a shift in the molecular weight distribution of RNA molecules captured within the brush towards large polynucleotides (>100 bp; Fig. 1d). Therefore, it is reasonable to propose that entropy gain is a driving force for such competitive adsorption, although local charge density and brush density may lead to kinetically-controlled competition. However, competitive binding

is clearly regulated by the molecular architecture of competitors, with large macromolecules displaying high charge densities associated with high adsorption rate constants $k_A$ and very rapidly displacing oligonucleotides to be delivered (Fig. 3). This is in agreement with the known ability of strong polyelectrolytes such as sulfated glycosaminoglycans to disrupt polyplexes.

In this context, our results indicate that molecular crowding at the brush surface and the molecular weight of competing electrolytes are important regulators of competitive desorption of small RNA. In our co-competition assay, with heparin and albumin at different concentrations (Supplementary Fig. 6), a reduction in the kinetics of desorption can be clearly seen. In addition, plotting $\Gamma_\infty$, the ultimate surface density of retained RNA (Equation 11), as a function of electrolyte concentration indicates the occurrence of plateaux for weak polyelectrolytes, whereas a linear relationship is observed within the range of concentrations tested for competitions with the small molecule glutathione (Supplementary Fig. 18).

Approximating the additivity of competitive desorption in the model proposed, it is possible to predict the extent of desorption that would be observed based on individual competitive binding, with the extent of desorption observed from cell lysate. This can be done using Eq. 2, analogous to Supplementary Equation 11 for additive competitive desorption.

$$\Gamma = \frac{k_{app1} C_S [Br]_0}{k_{app3}} - \sum_i \frac{k_{A_i}}{k_{app3}} \left[ A_{i_{bulk}} \right] = \frac{k_{app1} C_S [Br]_0}{k_{app3}} - \sum_i \frac{k_{A_i}}{k_{app3}\, \gamma_{Ai}} a_{Ai}$$

(2)

Where $i$ indicates the different electrolytes competing, $a_{Ai}$ is the molar thermodynamic activity of electrolytes $Ai$ and $\gamma_{Ai}$ is their molar activity coefficient.

Considering the approximated concentrations of proteins (200 mg/mL[32], assuming BSA can be used as model of the physico-chemistry of average cytosolic proteins), RNA (1 mg/mL[33]), free glycosaminoglycans (100 µg/mL, see 'Discussion' above) and small molecules (assuming only glutathione, at a concentration of 10 mM[28]), Eq. 2 predicts that the residual surface density of small RNA after exchange with cytosolic components should be negative. This clearly contrasts with the 1.0 µg/cm² residual surface density of RNA that can be observed in competitive experiments in pure lysates (Fig. 2a). Although all competitive species will also compete with each other (limitation of the approximation made for Eq. 2), our results suggest that molecular crowding in the cytosol is modulating competitive binding and the desorption of RNA from polycationic complexes (glycosaminoglycans alone would account for higher degrees of dissociation, at a concentration of 100 µg/mL). This is in agreement with predicted effects of molecular crowding on weakly attractive complexes[37], compared to non-interacting systems[38]. Such crowding is likely to result from the brush structure itself, as well as the macromolecular corona forming in the cytoplasm. However, cytosolic molecular crowding and macromolecule compartmentalisation (in particular in the case of glycosaminoglycans) are also likely to contribute significantly to the modulation of desorption kinetics and equilibria, for example via the concentration dependence of molar activity coefficients $\gamma_{Ai}$.

In turn, we propose that such competitive desorption, modulated by molecular crowding, should be an important factor directing the design of polycationic vectors, especially to ensure sustained therapeutic effect of delivered oligonucleotides. On the extra-cellular side, the role of GAGs and proteoglycans (at the cell surface) will dominate RNA displacement and may result in full release of RNA prior to cellular delivery. In contrast, in the cytoplasm, we find that exchanges are dominated by RNA and

protein complexes associated with translation, endosomal components and cytoskeleton-mediated cellular trafficking. Overall, stable RNA binding can help retarding release both extra- and intra-cellularly, as demonstrated in our study. However, other strategies could be developed, based on the model proposed. In particular, the design of the molecular crowding of polycationic vectors could allow to modulate competitive binding. In this respect, polymer brushes are well suited for the engineering of molecular crowding, as it can be simply achieved via the control of grafting density, as well as at the introduction of hierarchical architectures, such as block copolymers or mixed polymer brushes.

The identification of the composition of the polycationic vector-cytoplasmic interactome may have important implications for the design of gene delivery vectors and the understanding of their impact on cytotoxicity and off-target effects. Indeed, beyond the implication of RNA release via exchange with cytoplasmic RNA and associated macromolecules, an important pool of proteins associated with translation were identified in our assay. Amongst these, several ribosomal proteins (e.g. 60S acidic ribosomal protein P2 and 40S ribosomal protein S21)[39], proteins regulating the binding of mRNA to ribosomes and the initiation of their translation (such as eukaryotic translation initiation factors 4B and 4H)[40] and proteins associated with direct RNA binding and splicing (e.g. Y-box-binding protein 1)[41] were identified. How such binding impacts on the regulation of translation remains unknown, but we propose that it could underly a mechanism resulting in off-target effects of transfections, irrespective of the type and sequence of the genetic material to be delivered. In addition, the transcription co-activator YAP1, playing an important role in the regulation of the Hippo pathway and mechanotransduction[42], was identified as part of the proteome associated with both PDMAEMA and PMETAC brushes. How binding of YAP1 to polycationic vectors may impact on its regulation of transcriptional activity is unknown, but could impact downstream pathways and cell phenotype. Other proteins identified in our assay were associated with stress response (e.g. heme-binding protein 2, E3 ubiquitin-protein ligase PPP1R11) and could underly cytotoxic response typically associated with transfection.

Comparison of the cytosolic proteome associated with PDMAEMA and PMETAC brushes revealed remarkable similarities (Fig. 1a and Supplementary Fig. 1). Not surprisingly considering the strong electrolyte character of PMETAC, a broader pool of proteins was associated with these brushes, compared to PDMAEMA (87 compared to 50, respectively). However, the 3 main classes of proteins identified within the cationic vector proteome were detected in both types of brushes. The fourth pool of other cytosolic proteins was more restricted in the case of PDMAEMA brushes (only 6 out of 17 detected in PMETAC), perhaps reflecting the reduced charge densities associated with these brushes, and corresponding reduction of expected binding with weakly acidic proteins. Other major differences observed between the two types of proteome were attributed to the presence of 4 histone 2B proteins (potentially resulting from RNA/DNA binding to the strong polyelectrolyte) and keratins and microtubule associated proteins (potentially reflecting the limited acidity of the former, better able to bind PMETAC, and the natural association of the latter with positively charged residues typical of nuclear localisation factors) in the case of PMETAC. However, most ribosome associated and RNA binding proteins, as well as clathrins were observed in both proteomes, indicating that beyond differences in the binding and retention of RNA and other oligonucleotides, these two types of polyelectrolyte brushes may be expected to behave similarly in terms of transfection mechanisms.

Overall, in addition to potential regulation of cytoplasmic molecular signalling and biochemistry, understanding the polycationic vector-cytoplasmic interactome may also enable the improved control of their intracellular fate, for example limiting their association with proteins targeting them for lysosomal degradation and exocytosis. In this respect, the role of vector structure and physico-chemistry on the cytoplasmic interactome remains to be established. For example, harnessing the ability to control polymer brush architectures to present bioinert[18] and potentially biofunctionalised outer blocks, modulating cytoplasmic interactions and localisation. The sustained release from resulting vectors may enable the knockdown of target proteins with efficiencies comparable to those achieved with shRNA strategies, but without the requirement of viral transfection. For example, the level of knockdown efficiency achieved using cationic polymer brushes would allow sustained long term knockdown via one single non-viral transfection prior to each passage. Improving sustained delivery of RNA and the stability of associated vectors and cargos may also find application in RNA vaccine design, in which such stability is an important consideration for rapid deployment, but also to potentially reduce the dose of RNA required for each injection. Overall, the regulation of competitive displacement will also enable the wider applicability of RNA delivery from cationic polymeric vectors, for tissue engineering, systemic delivery and vaccination applications.

## Methods

**Materials**. 2-Dimethylaminoethyl methacrylate (DMAEMA), copper chloride (Cu(I)Cl), copper bromide (Cu(II)Br$_2$), 2,2'-bipyridyl (bpy), anhydrous toluene, triethylamine (Et$_3$N), methyl iodide, ethyl α-bromoisobutyrate initiator, bromoacetic acid and vinyl acetate were purchased from Sigma-Aldrich and used as received. All chemicals and solvents were analytical grades unless otherwise stated. Cu(I)Cl was kept under vacuum in a desiccator until use. Silicon wafers (100 mm diameter, <100> orientation, polished on one side/reverse etched) were purchased from Compart Technology Ltd and cleaned in a Plasma System Zepto from Diener Electronic, for 10 min in air. Silica particles (unfunctionalised) were purchased from Bangs Laboratories (supplied as powder, mean diameters of 300 nm and 2–3 μm). The silane initiator, (3-trimethoxysilyl)propyl 2-bromo-2-methylpropionate was purchased from Gelest. Surface plasmon resonance (SPR) chips (10 × 12 × 0.3 mm) were purchased from Ssens. Triton X-100, gelatin, phalloidin–tetramethylrhodamine B isothiocyanate, PFA (paraformaldehyde), DAPI (4,6-diamidino-2-phenylindole), phosphate-buffered saline (PBS, 150 mM), potassium chloride, sodium chloride, sodium orthovanadate, sodium fluoride, β-glycerol phosphate, disodium pyrophosphate, L-arginine, L-histidine, L-lysine, chondroitin sulfate, heparan sulfate, hyaluronic acid, agarose gel powder, ethidium bromide, bovine serum albumin, negative control siRNA (MISSION® siRNA Universal Negative Control), Cy5 fluorescent labelled siRNA (MISSION® siRNA Fluorescent Universal Negative Control Cy5), and 6-FAM fluorescent labelled siRNA (MISSION® siRNA Fluorescent Universal Negative Control, 6-FAM) were purchased from Sigma Aldrich. Dulbecco's modified eagle medium (DMEM) medium, OPTI-MEM™ medium, foetal bovine serum (FBS), trypsin, versene, penicillin-streptomycin, L-glutamine, hoechst 33342, live/dead assay kit and LysoTracker® Red DND-99 were from Thermo-Fisher. Collagen type I was from BD Bioscience. GFP siRNA (target sequence CGG CAA GCT GAC CCT GAA GTT CAT) was purchased from Qiagen®.

### Synthesis and characterisation of polymer brush coated silica nano- and microparticles

*Synthesis of PDMAEMA brushes from silica particles*. Silica particles (microparticles with average size of 3 μm, and nanoparticles with average size of 300 nm) were functionalised with PDMAEMA brushes. Briefly, 200 mg of silica particles were washed/suspended with anhydrous toluene (4 mL) three times, under inert atmosphere. The particles were then redispersed in anhydrous toluene. Anhydrous triethylamine (200 μL, Et$_3$N, distilled over potassium hydroxide) and the silane initiator ((3-trimethoxysilyl)propyl 2-bromo-2-methylpropionate, 40 μL) were added to the resulting suspension and the mixture was allowed to react overnight with continuous agitation. Silane initiator-coated particles were washed three times with ethanol and stored in 10 mL water/ethanol (4/1 v/v), at 4 °C, until used for polymerisation. DMAEMA (6.6 g, 42 mmol), bpy (320 mg, 2.05 mmol) and CuBr$_2$ (80 mmol) were dissolved in water/ethanol 4/1 (v/v; 20 mL). The resulting mixture was degassed for 30 min with argon bubbling whilst stirring prior to the addition of and CuCl (0.082 g, 828 μmol) and further degassing. Polymerisation was allowed to proceed under argon at RT for 20 min to achieve a PDMAEMA brush thickness of 30 nm. To stop the polymerization, the reaction mixture was diluted with deionised

water and air was bubble through, until the colour changed from dark brown to blue. PDMAEMA-grafted silica particles were separated via centrifugation, at 3100 × g for 10 min, successively washed with water and ethanol (each time with centrifugation), to remove the catalyst and residual monomer. Resulting particles were stored in after drying, in a fridge.

*Quaternisation of PDMAEMA-brush functionalised silica particles*. Particles functionalised with PDMAEMA brushes (200 mg) were suspended in DMSO (10 mL) together with iodomethane or 1-acetoxyethyl-2-bromoacetate (100 mM). The resulting mixture was allowed to react at room temperature for 24 h, with stirring. Particles were centrifuged at 3100 × g for 10 min, washed with water and ethanol (at least three times each) and stored at 4 °C.

*Characterisation of polymer brush-coated silica nanoparticles*. The size and ζ-potential of quaternised PDMAEMA coated silica nanoparticles were measured using a Malvern zetasizer nano ZS. Samples (1 mg/mL dispersed in PBS for size and diluted PBS (10×) for ζ-potential) were repeatedly sonicated for 10 min, followed by gentle shaking, until homogenously dispersed. Each sample was characterised in triplicates (measurements carried out at 25 °C). Attenuated total reflection Fourier-transform infrared spectroscopy (ATR-FTIR) was carried out using a Bruker Tensor 27 with an MCT detector (liquid N$_2$ cooled). Acquisition of spectra was at a resolution of 4 cm$^{-1}$ (128 scans per run in total), using OPUS 8.5.29. Thermogravimetric analysis (TGA) was carried out in air using a TA Instrument Q500 and the TA Q Series Version 2.5.0.256 software. All samples were dried under vacuum at room temperature prior to analysis via TGA. Heating of samples was initiated at room temperature, up to 1000 °C, with a heating rate of 10 °C/min. The weight loss of polymer on silica nanoparticles was determined at 900 °C.

*pH responsiveness of CS-PMETAC brushes*. CS-PMETAC-functionalised silica nanoparticles were dispersed in PBS (pH adjusted to 7.4 and pH 5, respectively) and incubated at room temperature for periods of time ranging from 30 to 240 min. Following this incubation, particles were washed and resuspended in PBS (pH 7.4), followed by characterisation via dynamic light scattering (DLS) and electrophoretic dynamic light scattering, using methods described above.

### Synthesis and characterisation of polymer brushes grown from silicon substrates and free PDMAEMA

*Synthesis of 1-acetoxyethyl-2-bromoacetate*. 1-acetoxyethyl-2-bromoacetate was synthesised following protocols previously reported[43]. In brief, 4.168 g (30 mmol, 1.5 eq) of bromoacetic acid were added to 1.721 g (20 mmol, 1 eq) vinyl acetate in a round-bottomed flask (100 mL). The resulting mixture was brought to 95 °C and kept at this temperature for 19 h until all vinyl acetate had reacted. The resulting mixture was left to cool down. Purification was carried out via chromatography (petroleum ether/ethyl acetate 6/1). The desired product was obtained as a yellowish liquid (yield 40 %). $^1$H NMR(400 Hz, CDCl$_3$): δ 6.79 (q, J = 16 Hz, 1H), 3.75 (s, 2H), 2.00 (s, 3H), 1.43 (d, J = 4 Hz, 3H).

*Synthesis of free PDMAEMA polymer*. Free PDMAEMA polymer was synthesised via ATRP following a method adapted from previous reports[18,44]. DMAEMA (15 g, 95.5 mmol, 500 eq), ethyl α-bromoisobutyrate initiator (EBiB, 0.073 g, 0.374 mmol, 2 eq) and 2,2'-bipyridyl ligand (bpy, 0.0297 g, 0.190 mmol, 1.01 eq) were mixed in ethanol and water (10 mL, 1/4 v/v) in Ia 100 mL round bottom flask. Bubbling of nitrogen gas through the resulting mixture for 30 min (RT) allowed the removal of oxygen. Copper (I) chloride (0.0186 g, 0.188 mmol, 1 eq) was then added to the mixture. The reaction was heated to 50 °C. A positive pressure of nitrogen was kept throughout the polymerisation. After 5 h, the reaction mixture was cooled to room temperature and exposed to air. The solution was dialyzed (2 L beaker, deionised water) using a Spectra/Por regenerated cellulose membrane with a molecular weight cut off of 1000 Da. The resulting aqueous solution was then lyophilised to yield the final polymer as an off white solid. $^1$H NMR(400 Hz, D$_2$O): δ 4.14 (s, 2H), 2.74 (s, 2H), 2.33 (d, J = 8 Hz, 6H), 1.80–1.71 (m, 2H), 1.12–0.89 (m, 3H).

*Quaternisation of free PDMAEMA with 1-acetoxyethyl-2-bromoacetate*. PDMAEMA (500 mg, 3.2 mmol, 1 eq) and 1-acetoxyethyl-2-bromoacetate (1.075 g, 4.8 mmol, 1.5 eq) were added to 5 mL DMF in a round bottom flask (25 mL). The resulting mixture was heated to 70 °C and stirred at this temperature overnight. The resulting material was precipitated in ethanol (cooled) twice. The resulting powder was dried in vacuum. $^1$H NMR(400 Hz, D$_2$O): δ 6.92 (d, J = 20 Hz, 1H), 4.69–4.28 (m, 2H), 4.24–82 (s, 2H), 3.63–3.51 (m, 2H), 3.45 (s, 2H), 3.33 (s, 6H), 2.98 (s, 6H), 2.11 (s, 3H), 1.97 (m, 2H), 1.52 (m, 3H), 1.14–0.97 (m, 3H). Note that partial cleavage of charge shifting groups was observed in water, accounting for the reduced integration of corresponding peaks (at 6.92, 3.63–3.51, 3.45, 2.11 and 1.52 ppm), but full functionalisation and disappearance of the methyl amine peaks of PDMAEMA (2.33 ppm) was observed.

*Quaternisation of PDMAEMA brush with 1-acetoxyethyl-2-bromoacetate and methyl iodide*. PDMAEMA brushes (30 nm) were grown from silicon substrates as describe above. The resulting PDMAEMA brush-coated substrates were incubated

in 1-acetoxyethyl-2-bromoacetate or methyl iodide solutions (in DMF, 100 mM), for 12 h. The resulting quaternised polymer brushes, namely PMETAC for methyl iodide quaternisation, CS-PMETAC for brushes reacted with 1-acetoxyethyl-2-bromoacetate, were washed with copious amounts of ethanol and dried in a stream of nitrogen. Characterisation of resulting coatings was carried out by ellipsometry.

*pH responsiveness of CS-PMETAC.* CS-PMETAC brush-coated silicon substrates were placed in buffers of PBS with pH of 7.4 or adjusted to 5.0. Substrates were left to incubate at room temperature for 4 h. Dry brush thicknesses were characterised via ellipsometry prior and directly following this incubation.

*Wet ellipsometry measurement for brush swelling at different pH.* Polymer brush-functionalised substrates (1 × 3 cm); PDMAEMA, PMETAC and CS-PMETAC) were mounted in the liquid flow chamber of the ellipsometer (α-SE instrument from J.A. Woollam at an incidence angle of 70º, using the Complete EASE software). For dry samples, the model used for fitting of the data consisted in a silicon substrate/Cauchy bilayer. Substrates were mounted in the flow chamber (fitted with quartz windows normal to the beam path). Samples were dried in an argon flow (within the chamber) maintained for at least 10 min. The dry brush thickness was then measured, under argon. Following this measurement, 4 mL deionised water were injected in the chamber and brushes were allowed to equilibrate for 5 min before measuring brush swollen thicknesses. Similarly, brush swelling was subsequently measured in 150 Mm NaCl, pH 7.4 PBS, pH 5 150 Mm NaCl and pH 9 150 Mm NaCl.

**SiRNA binding on quaternised PDMAEMA brushes.** PDMAEMA, PMETAC and CS-PMETAC brush-functionalised chips for surface plasmon resonance (SPR) were prepared according to methodologies reported in the literature[18]. SPR enabled the quantification of the adsorption of double-stranded RNA oligonucleotides (22 bp siRNA) to the surface of corresponding polymer brushes. Measurements were carried out using a Biacore 3000 instrument (using the Biacore X Control Software). Polymer brushes were grown from SPR chips (following protocols described above, using ω-mercaptoundecylbromoisobutyrate as initiator) and then mounted on the substrate holder. Mounted chips were docked, primed with PBS and equilibrated in PBS at a flow rate of 10 μL/min, until a stable baseline was obtained. The siRNA solution (50 μL, 10 μg/mL) was injected and binding was monitored as the shift in response unit. Following completion of the injection, substrates were washed with PBS, with a continuous flow (10 μL/min). Adsorption levels were quantified after stabilisation of the signal. For acidic degradation experiments, the buffer was then switched to acidic PBS (pH 5.5) until the signal was stabilised again. All measurements were carried out in triplicates (three separate chips, freshly prepared).

**Formation of PDMAEMA brush-siRNA polyplexes and characterisation of release kinetics, competitive binding and co-adsorption.** PDMAEMA grafted silica microparticles were dispersed in RNAse free water to make stock solutions of 10 mg/mL. 200 μL of SiO2-PDMAEMA microparticle dispersion was mixed (vortexing for 30 s) with an equal volume of Cy5 tagged siRNA (22 bp) in RNAse free water at an N/P ratio of 10 (final RNA concentration of 20 μg/mL) and allowed to assemble for 30 min at room temperature. The polyplexes were then centrifuged at 3100 × g for 10 min. The supernatant was removed and washed three times with PBS pH 7.4. The pellet was redispersed in phosphate buffer saline (PBS), pH 7.4. The polyplex dispersion in PBS was used as negative control (limited release). To characterise the kinetics of release in various conditions, the polyplex pellets as prepared above was dispersed in the corresponding test solutions (containing different competitors dissolved in PBS), and imaged using confocal microscopy. Mean fluorescence intensities of at least 50 microparticles were quantified from 3 images at each time point. After background correction (subtracting the intensity of areas not containing any particle), mean fluorescence intensities were calculated for each time points and averages of three separate experiments were combined. A summary of the different aqueous solutions used in this assay include: PBS with pH adjusted to 5.5 and 10.0 and PBS solutions of chondroitin sulfate, heparin, hyaluronic acid, amino acids (arginine, histidine), bovine serum albumin and glutathione reductase (GSH) at pH 7.4, as well as aqueous solutions of KCl and NaCl at different ionic strength (pH of 7.4). In addition, siRNA solutions for release studies were prepared by diluting negative control siRNA vials (10 nM) with 50 μL of RNAse free PBS, pH 7.4. For co-adsorption experiments (Supplementary Fig. 4), following adsorption of 6-FAM-tagged RNA onto PDMAEMA brush-microparticles, the brush complexes were washed in PBS and incubated in BSA and poly(L-lysine) solutions (both from Sigma; PLL is 30–70 kDa; both tagged with Alexa Fluor™ 594 N-hydroxy succinimidyl ester from Thermo Fisher Scientific).

For competitive binding assays in mixed solutions of heparin and BSA, the following protocol was applied. PDMAEMA grafted silica microparticles were dispersed in RNAse free water to make stock solutions of 10 mg/mL. 200 μL of SiO2-PDMAEMA microparticle dispersion was mixed with an equal volume of cy5 tagged siRNA (22 bp) in RNAse free water at an N/P ratio of 10 (final RNA concentration of 20 μg/mL) and allowed to assemble for 30 min at room temperature. The polyplexes were then centrifuged at 3100 × g for 10 min. The

supernatant was removed and washed three times with PBS (pH 7.4). The pellet was redispersed in phosphate buffer saline (PBS), in a small volume (10 μL). To characterise the kinetics of release in mixed solutions of heparin and BSA, the polyplex pellets as prepared above was deposited in Ibidi wells and incubated with 240 μL of BSA (40, 100, 125, 150, 200, 250 mg/mL), and then another 250 μL of heparin (40 μg/mL) was added, and imaged using epifluorescence microscopy. Mean fluorescence intensities of at least 50 microparticles were quantified from 3 images at each time point. After background correction (subtracting the intensity of areas not containing any particle), mean fluorescence intensities were calculated for each time point and averages of three separate experiments were calculated.

**Preparation of whole-cell lysate for in vitro release studies.** Whole-cell lysates of human fibroblasts (HCA2) were prepared, for siRNA release studies, as per previously reported protocol with slight modifications[45]. Briefly, HCA2 cells were grown in T75 flasks, at 37 °C in 5% CO2 atmosphere, using DMEM medium supplemented with 10% fetal bovine serum (FBS), 1% glutamine and 1% antibiotics (Penicillin/streptomycin). Fully confluent cells from each flask were detached by trypsinization. Cells were then pelleted by centrifugation at 300 × g for 5 min. The supernatant was removed and the cell pellets were washed with ice-cold PBS three times, followed by aspirating the supernatant. The cell pellets were subjected to three freeze thaw cycles by dipping in liquid nitrogen and thawing on ice for a few minutes. The pellets were then subjected to bath sonication for 30 s, and centrifuged at 10,621 × g for 20 min. Cell lysate supernatants were collected for performing release studies.

**Horizontal gel electrophoresis.** Binding of nucleic acids to the polymer brush grafted silica particles was characterised by performing agarose gel electrophoresis. The polymer brush particle complexes (polymer brush -siRNA complexes at N/P ratio of 10) were incubated with cell lysates for 30 min and separated by centrifugation at 3100 × g for 10 min. The pellets were washed repeatedly with PBS, pH 7.4, and the adsorbed macromolecules were desorbed via incubation in a 2 M NaCl aqueous solution. The nucleic acid solutions were mixed with gel loading buffer (5/1 v/v), and loaded on to 1.5% agarose gel containing ethidium bromide (0.5 mg/mL of gel) in Tris buffer. Ladder with a molecular weight range of 100–3000 bp was used as molecular weight markers. Electrophoresis was carried out for 60 min, at 70 V. Bands within the gels were observed using a UV illuminator, and photographs were taken using a Molecular Imager system (Gel Doc XR, BIO RAD). Full scan blots can be found with the source data files.

**Preparation of cytosolic fractions for proteomics analysis.** HCA2 cells were cultured in T75 flasks, at 37 °C in 5% CO2 atmosphere, using DMEM medium supplemented with 10% fetal bovine serum (FBS), 1% glutamine and 1% antibiotics (Penicillin/streptomycin). Cell detachment was achieved by aspirating the medium, washing with ice-cold PBS containing a protease inhibitor once and adding an appropriate volume (2 mL for a 75 mL flask) of ice-cold fractionation buffer (HEPES, 20 mM; KCl, 10 mM; MgCl2, 2 mM; EDTA, 1 mM; EGTA, 1 mM; just before use, 1 mM DTT and a protease inhibitor cocktail (protease inhibitor cocktail III from Abcam, 50 μL for 100 mL of buffer; full detail of composition of the stock solution: AEBSF hydrochloride, serine protease inhibitor, 20 mM; Bestatin, aminopeptidase B and leucine aminopeptidase inhibitor, 1.7 mM; E-64, cysteine protease inhibitor, 0.2 mM; EDTA disodium salt, metalloprotease inhibitor, 85 mM; Pepstatin A, aspartic protease inhibitor, 2 mM) were added). Flasks, samples and buffers were kept on ice throughout the protocol and centrifugation was carried out at 4 °C. Using a 1 mL syringe, cell suspensions were passed through a 27-gauge needle 10 times to mechanically lyse cells. The resulting suspension was left on ice for 20 min. The samples were then centrifuged (720 × g) for 5 min. Pellet were discarded and the supernatants were isolated, containing cytoplasm, membrane and mitochondria. Supernatants were transferred into a fresh low adhesions tube and kept on ice, before centrifugation (10,000 × g for 5 min). The pellets containing mitochondria were discarded before transferring the supernatant into a fresh tube, on ice. This contained the cytoplasm and membrane fraction. The supernatant was further centrifuged in an ultracentrifuge at 100,000 × g for 1 h. This resulted in cytosolic fractions that were used for proteomics analysis.

**Polymer brush-cell lysate/cytosolic fraction interactions and sample preparation for proteomics analysis.** Human fibroblasts cells (HCA2) were cultured in T75 flasks as described earlier and the media were aspirated after cells reached confluency. Appropriate volume (2 mL for a 75 mL flask) of ice-cold PBS containing protease and phosphatase inhibitors (20 μL NaF and 100 μL Na3VO4 per 10 mL of PBS, pH 7.4) were introduced onto the cell layer in the T75 flask. Flasks were kept on ice, the PBS was aspirated and the process was repeated twice. 0.5 mL of lysis buffer (8 M urea in 20 mM HEPES, pH 8.0; Na3VO4, 100 mM; NaF, 500 mM; β-glycerol phosphate, 1 M; Na2H2P2O7, 250 mM) was added to the cells followed by gentle scrapping using a cell scrapper. The cell suspension was lysed by probe sonication at 50% intensity for 15 s, twice, with a 10 s rest period in between. The vial containing the cell suspension was immersed in ice throughout the sonication process to ensure that heat generated by sonication did not adversely affect the phosphoprotein content. The cell suspension was centrifuged at

$17,950 \times g$ for 10 min at 5 °C and the supernatant was transferred to a 1.5 mL Eppendorf Protein Lo-bind tube. A portion of this cell lysate was incubated with SiO$_2$-PDMAEMA brushes, at a concentration of 5 mg/mL, for 30 min and the pellet was recovered by centrifugation at $3100 \times g$ for 10 min. The pellets were washed with ice-cold PBS, pH 7.4 by centrifugation at 4 °C. Frozen dried pelleted beads were thawed in ice and suspended in 280 µL freshly prepared 25 mM ammonium bicarbonate solution. 500 ng of sequence grade trypsin (Thermo Scientific, Cat log no 13454189) was added into each sample and incubated overnight at 37 °C with shaking at 1400 rpm. The digested peptides were desalted using C18+ carbon top tips (Glygen Corporation, TT2MC18.96) and eluted with 70% acetonitrile (ACN) with 0.1% formic acid and dried in a SpeedVac vacuum concentrator.

For interaction of SiO$_2$-brush nanoparticles with cytosolic fractions (prepared as detailed above, by sequential centrifugation), we incubated SiO$_2$-PDMAEMA and SiO$_2$-PMETAC nanoparticles at a concentration of 5 mg/mL in undiluted cytosolic fractions, for 30 min. The pellets were recovered by centrifugation at $3100 \times g$ for 10 min. The samples were then processed for proteomics analysis, as described above.

**Whole-cell lysate preparation and sample preparation for proteomics**. In solution digestion and mass spectrometry analysis was carried out following a protocol described in the literature[46]. Protein content within the lysates was quantified in urea buffer (8 M in 20 mM HEPES pH 8.0) using a BCA protein assay. 50 µg of protein for each sample were diluted to a final volume of 300 µL with 8 M urea buffer. Protein solutions were sequentially incubated with 10 µL of DTT (1 M) and 4 µL of iodoacetamide (IAM, 15 mM) for 1 h and 30 min, respectively at room temperature, whilst shaking. Urea in samples was diluted to 2 M by adding 900 µL of HEPES buffer (20 mM pH 8.0) prior to adding 90 µL immobilised trypsin beads (Life Technologies Ltd, cat. 10066173). Samples were kept overnight at 37 °C with shaking. Trypsin beads were removed by centrifugation and peptide solutions were desalted using C18+ carbon top tips (Glygen Corporation, TT2MC18.96). The tips were activated with 200 µL of acetonitrile (ACN), and equilibrated with 200 µL of wash solution (1% ACN, 0.1% TFA). Samples were loaded in the tips and washed with 200 µL of washing solution. Peptides were eluted with 200 µL of elution solution 1 (70% ACN, 0.1%TFA) and dried in a SpeedVac vacuum concentrator.

**Mass spectrometry**. Dried peptides were dissolved in 0.1% TFA and analysed by nanoflow ultimate 3000 RSL nano instrument, coupled on-line to a Q Exactive plus mass spectrometer (Thermo Fisher Scientific). Gradient elution was from 3 to 35% buffer B in 120 min at a flow rate 250 nL/min with buffer A being used to balance the mobile phase (buffer A was 0.1% formic acid in water and B was 0.1% formic acid in ACN). The mass spectrometer was controlled by the Xcalibur software (version 4.0) and operated in the positive mode. The spray voltage was 1.95 kV and the capillary temperature was set to 255 °C. The Q-Exactive plus was operated in data dependent mode with one survey MS scan followed by 15 MS/MS scans. The full scans were acquired in the mass analyser at 375–1500 $m/z$ with the resolution of 70,000, and the MS/MS scans were obtained with a resolution of 17,500.

**Peptide and protein identification and quantification**. For peptide identification, MS raw files were converted into Mascot Generic Format using Mascot Distiller (version 2.7.1) and searched against the SwissProt database (release September 2019) restricted to human entries using the Mascot search daemon (version 2.6.0) with a FDR of ~1% and restricted to the human entries. Allowed mass windows were 10 ppm and 25 mmu for parent and fragment mass to charge values, respectively. Variable modifications included in searches were oxidation of methionine, pyro-glu (N-term) and phosphorylation of serine, threonine and tyrosine. Peptides with an expectation value <0.05 were considered for further analysis. The mascot result (DAT) files were extracted into excel files. For peptide quantification Pescal Software was used to construct extracted ion chromatograms for all the identified peptides across all conditions. For protein quantification, an in house developed script was used to sum the intensities of all peptides comprised in the same protein. Only proteins with at least 2 different peptides and a score >40 were considered for further analysis. Cluster analysis was performed via ClustVis[47]. Functional cluster analysis was performed via DAVID Bioinformatics Resources 6.8[48].

**Transfection with polymer brush-coated silica nanoparticles**. HaCaT-GFP cells (expressing actin-GFP, generated by transfection with linearized plasmids for EGFP-actin[18,49]) were seeded at densities of 25, 12.5, 6.25 and 3.125 k/well) on collagen-treated glass coverslips, in 24-well plates. Cells were allowed to adhere on these substrates for 24 h. Transfections were carried out with GFP siRNA at a concentration of 50 nM/well. 100 µL of polymer brush-coated nanoparticles complexing GFP siRNA were formulated (N/P = 10 and 20) OPTI-MEM medium (serum free). The DMEM medium was removed from each well and remaining adhered cells were washed twice with pre-warmed OPTI-MEM medium. Finally, after a final aspiration of medium, 400 µL of OPTI-MEM was added. 100 µL siRNA complex (final concentration indicated above) was then slowly added to each well, followed by gentle mixing. Incubation of the resulting cultures for 4 h was allowed

prior to exchange of the medium with 500 µL of regular DMEM medium. Further culture was allowed to continue for 1–4 days, depending on the starting cell density (1 d, 25 k; 2 d, 12.5 k; 3 d, 6.25 k; 4 d, 3.125 k). In the case of longer-term transfections (10 d), the starting concentration of HaCaT-GFP cells was initially 3.125 k/well, but at day 5 of culture cells were trypsinised and re-seeded on a new coverslip freshly coated with collagen at a density of 3.125 k cells/well. As positive/negative controls, for all time points, Lipofectamine® 2000 complexed with GFP siRNA or negative control (NC) siRNA, respectively (following the protocol recommended by the manufacturer, at a final siRNA concentration of 50 nM/well). At the desired time point, transfected cells were washed with PBS (three times), fixed in paraformaldehyde (PFA, 4%, 10 min) and permeabilised with Triton X-100 (0.2%, 5 min). Cells were then stained with TRITC-phalloidin (1:1000) and DAPI (4,6-diamidino-2-phenylindole, 1:1000) in blocking buffer (10% FBS and 0.25% gelatin from cold water fish skin, Sigma-Aldrich) and kept at room temperature for 1 h. Coverslips with fixed cells were mounted on glass slides before imaging with a Leica DMI8, using the Leica Application Suite X (3.4.2.18368) epifluorescence microscope. Transfection efficiency was quantified using Image J.

**Quantification of cell viability in the presence of polymer brush-coated silica nanoparticles**. HaCaT cells were seeded at different densities and transfected as described above. Cell viability was quantified via a live/dead assay. At the desired time point (corresponding to different cell densities, as per the transfection assay), cells were incubated in 500 µL DMEM medium containing 4 mM calcein AM, 2 mM ethidium homodimer and Hoechst 33342 (staining all cell nuclei) for 30 min, at 37 °C (in the incubator). Fluorescence imaging was carried out as shortly as possible afterwards. Live/dead cells and total cell densities were counted on each image, using ImageJ.

**Labelled silica nanoparticles and siRNA co-localisation study in HaCaT cells**. HaCaT cells were seeded on collagen-coated glass substrates (coverslips), at a density of 12.5 k/well, in 24-well plates. Cells were allowed to adhere and spread for 24 h, prior to transfection. Labelled PDMAEMA-functionalised nanoparticles were generated using a macroinitiator approach. Polyelectrolyte multilayers containing the macroinitiator were generated, introducing a fluorescent conjugated polyelectrolyte as stable label, according to a previously reported methodology[19]. This allowed polymer brushes with comparable grafting densities to those generated from initiator silane or thiol monolayers to be achieved. Quaternisation of the resulting PDMAEMA brushes was carried as described above. 6-FAM labelled siRNA (sigma Aldrich, MISSION® siRNA Fluorescent Universal Negative Control, 6-FAM) was used to track siRNA binding and release upon cytoplasmic entry. A final siRNA concentration of 50 nM/well was used. 100 µL of PDMAEMA-brush functionalised nanoparticles complexing 6-FAM siRNA (and PMETAC and CS-PMETAC complexes) were prepared at N/P=20, in serum free OPTI-MEM medium. After removal of the medium, cells were washed twice with pre-warmed serum free OPTI-MEM medium. 400 µL of OPTI-MEM medium was added and 100 µL of siRNA complex suspension was slowly added in each well, followed by gentle agitation. Incubation was allowed fo 4 h, in the incubator. The medium was then replaced with 500 µL of DMEM medium, and culture was allowed to continue for 2 days. Finally, cells were stained with LysoTracker® Red DND-99 at 50 nM for 1 h at 37 °C and fixed in paraformaldehyde (PFA, 4%, 10 min) before imaging with a Zeiss LSM710 Confocal and Elyra PS.1 Superresolution microscope (using Zen 2012 sp5). Images were analysed and quantified with Image J.

*Protocol for particle/RNA/endosome co-colocalisation analysis*. Images were analysed with Image J. Images were split into separate colour channels. The channel corresponding to particles (blue channel) was converted into binary images (using Thresholding), in order to identify the position of all particles in each image. Particles aggregating were separated using the "Watershed" function. To quantify the co-localisation of different components, intensity measurement associated with each particle (in the particle/blue channel) were re-directed to the corresponding channel. The "Analyze Particles" function was used to identify all particles from the binary images and measurements were redirected to the corresponding channels, separately. Results were exported, retaining registry of the particle number for each dataset. This enable the tracking of particles in different channels for further data analysis. Finally, intensities were compared to determine whether RNA were retained at the particle surface and whether particles had escaped the endosome.

**Statistical analysis**. For statistical analysis, one-way ANOVA with Tukey test for posthoc analysis was used (in Origin 8). Significance was determined by $*P < 0.05$, $**P < 0.01$, $***P < 0.001$, n.s. not significant.

**Reporting summary**. Further information on research design is available in the Nature Research Reporting Summary linked to this article.

## Data availability
Supplementary discussion, additional proteomics data, cluster analysis, functional clustering analysis, additional competitive binding kinetics, [1]H NMR spectra, XPS, ellipsometry, DLS, FTIR, TGA, SPR, cell viability, additional fluorescence microscopy

data and statistical analysis summary tables are available in the Supplementary Information. Source datasets for all figures are available from the corresponding author and have been deposited in the QMUL repository. Proteomics data generated in this study have been deposited to the ProteomeXchange Consortium via the PRIDE partner repository with the dataset identifier PXD028512 and 10.6019/PXD028512. Source data are provided with this paper.

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

## Acknowledgements

We thank Dr. Vinothini Rajeeve for proteomics analysis, Dr Ruth Rose for help with gel electrophoresis and Dr. John Connelly for assistance with analysis of the proteomics data. This work was supported by China Scholarship Council (D.L., grant no 201406240022 and L.C., grant no 201806100218), the Commonwealth Secretariat Commission (A.M.R.; Commonwealth Rutherford Fellowship, INRF-2017-179), and from the European Research Council (J.E.G.; ProLiCell, 772462).

## Author contributions

A.A.M.R., D.L., L.C. and J.E.G. designed the research. A.M.R., D.L. and L.C. synthesised and characterised materials and carried out experiments. A.M.R., D.L., L.C. and J.E.G. carried out data analysis. A.M.R., D.L. and J.E.G.co-wrote the manuscript. All authors edited the manuscript and approved its final version.

## Competing interests

The authors declare no competing interests.
