## [Peer Review File · Nature Communications]

REVIEWER COMMENTS

Reviewer #1 (Remarks to the Author):

In their manuscript "Competitive Binding and Molecular Crowding Regulate the Cytoplasmic Interactome of Non-Viral Polymeric Gene Delivery Vectors" the authors show that the molecular structure and architecture of cationic polymeric vectors and the cytosolic molecular crowding modulate competitive binding and the long term release of RNA and associated transfection efficiency. The authors employed PDMAEMA brush-functionalised nanoparticles as polycationic gene delivery vectors and characterised the cytoplasmic proteome associated with the vectors. Their ideas and design are novel, and clearly different from previous studies that gene release from polymeric vectors were modulated by a variety of stimuli and processes, including light, exposure to glutathione and hydrolytic degradation of charge shifting systems. The authors' investigation provides a new idea and method for the study and design of siRNA polycationic delivery vectors. This paper will draw widespread interests from the investigators in the field of gene therapy and non-viral vectors.

However, there are some significant issues should be addressed.

1. The authors indicated that the high density of polymer brushes was found to impact not only the binding capacity of these coatings, but also their kinetics of adsorption, depending on the molecular weight of the macromolecules adsorbed. Does it have to do with charge of the macromolecules adsorbed?
2. The authors indicated that a range of molecules and macromolecules significantly contributed to oligonucleotide displacement from non-viral polycationic vectors. Please estimate and provide the individual contribution from molecules and macromolecule.
3. In Figure 5 G, H, the KD efficiency of Lipo group at day 4 is similar to that at day 10 (about 50%) , the KD efficiency of PMETAC-functionalised nanoparticles at day 4 is similar to that at day 10 (about 75%). The authors indicated that the modulation of RNA displacement in polymer brush-based vectors enabled the sustained delivery of siRNA and led to prolonged knock down in their model. Does Lipo enable the sustained delivery of siRNA and led to prolonged knock down ? Please add the detailed explanation.

Reviewer #2 (Remarks to the Author):

Detailed mechanism governing polyelectrolyte complex nanoparticle-mediated transfection process remains elusive. This manuscript presented evidence supporting the hypothesis that the RNA release is regulated by competition with macromolecules within the cytoplasm. However, multiple questions still remain that limit the potential impact of the findings. Some of the conclusions are overly extended without sufficient support from the data reported. On the whole, the manuscript requires significant more work to reach the quality and impact for the journal. Specifically, major concerns and issues need to be addressed include:

- The approach of using a proteomics analysis to screen out the cytoplasmic proteins with strong binding affinities to PDMAEMA brush-grafted nanoparticles is a plus. However, the analysis did not provide any findings that were not known previously or unexpected, particularly in light of the qualitative nature of the conclusion "RNA displacement is clearly regulated by the strength and charge density of the macromolecules competing, as well as their molecular weight."
- Another limitation of this work is that the second half of the paper has minimal connection with the results from the detailed proteomics analysis. The concept of charge conversion carrier has been reported previously. The confirmation of the hypothesis on RNA release was only performed with RNA displacement test. What is the utility of the sophisticated analysis completed for the design or validation of the polycationic carriers?
- Establishing the molecular mechanism that regulates cytosolic dissociation of polycationic gene delivery vectors is valuable. What may be the limitation of using a whole cell lysate which may

include all components inside cytosol, cellular vesicles and nucleus? The conclusion of this study seemed to suggest that RNA competition is the primary factor to consider for cell transfection process.

- The abstract does not capture the key findings of this paper and included only vague and qualitative statements.
- The finding of adsorption of high IP proteins is interesting. Unfortunately, there was not follow up; the explanation on these proteins adsorbed "indirectly" to polycationic brushes is handwaving at best.
- The statement on "polymer specific" adsorption is not sufficient. How would such a conclusion compromise the finding with a PDMAEMA brush-coated nanoparticles?

Reply to reviewers' comments

Reviewer 1

1. The authors indicated that the high density of polymer brushes was found to impact not only the binding capacity of these coatings, but also their kinetics of adsorption, depending on the molecular weight of the macromolecules adsorbed. Does it have to do with charge of the macromolecules adsorbed?

The reviewer is correct that the surface charge and charge density of macromolecules competing will have an impact on the ultimate surface density of macromolecules adsorbed, as well as the kinetics of adsorption. In the present manuscript, we show that competition is strongly dependent on the charge density of competing macromolecules, as well as their size (e.g. glutathione vs. other macromolecules) and quantify the associated adsorption rate constant k_A (Figure 3G). We find that k_A varies over orders of magnitude, depending on the size and charge density of corresponding molecules. In this respect, molecules such as hyaluronic acid, chondroitin sulfate and heparin not only display different densities of ionisable moieties, but also different degrees of ionisation as carboxylates are typically not completely deprotonated even at neutral pH in weak polyelectrolytes, whereas sulfate groups will be. This is more specifically discussed in our revised manuscript (p16, "However, competitive binding is clearly regulated by the molecular architecture of competitors...").

0. The authors indicated that a range of molecules and macromolecules significantly contributed to oligonucleotide displacement from non-viral polycationic vectors. Please estimate and provide the individual contribution from molecules and macromolecule.

Considering the huge diversity of molecules and macromolecules present in the cytosol, quantifying the contribution of each of these components is unrealistic. However, we have quantified the contribution of small charged molecules such as glutathione, which may be present at concentrations up to 10 mM, as well as that of macromolecules such as proteins (in particular with low IP such as BSA, present at concentrations in the range of 200 mg/mL), RNA (present at concentrations in the range of 1 mg/mL) and glycosaminoglycans (present at concentrations in the range of 100 μ g/mL). We propose that a realistic indicator for such comparison is the $k_A[A_{bulk}]$ factor (modulated by parameters controlling oligonucleotide adsorption and infiltration within brushes), which takes into account the competitor concentration and its adsorption rate constant. Figure 3G is presenting results extracted from our fits that quantify k_A for the different molecules investigated in our study. This figure was revised to present results of statistical analysis. This confirms the impact of individual molecules and draws some correlation between their size and charge density and their impact on RNA displacement. Based on this analysis, it is clear that glycosaminoglycans should contribute very significantly to desorption and that small molecules, at relevant physiological concentrations, are unlikely to do so. In addition, our analysis highlights that RNA molecules are likely to be the next important contributors to RNA displacement in the cytosol. Finally, the absence of total desorption in the extra-cellular environment, where glycosaminoglycans can be particularly high, and in the cytosol, often associated with lower GAG concentrations, indicates that molecular crowding and partitioning of these macromolecules (e.g. at cell membranes) play important roles in the long term stability of RNA/cationic polymer vector complexes. These points are discussed in our manuscript on p17, in the second paragraph ("Considering the approximated concentrations of proteins...").

3. In Figure 5 G, H, the KD efficiency of Lipo group at day 4 is similar to that at day 10 (about 50%), the KD efficiency of PMETAC-functionalised nanoparticles at day 4 is similar to that at day 10 (about 75%). The authors indicated that the modulation of RNA displacement in polymer brush-based vectors enabled the sustained delivery of siRNA and led to prolonged knock down in their model. Does Lipo enable the sustained delivery of siRNA and led to prolonged knock down? Please add the detailed explanation.

Our data indicates that Lipofectamine is resulting in an initial peak in KD efficiency (from day 1), followed by a decrease to a moderate KD level (after day 3). This then persists over the following week of incubation, at a lower level than what is observed for PDMAEMA brush coated nanoparticles. The retention of KD, although at a reduced level, may indicate that a pool of proteins (in our model, actin) is not recycled as fast and may persist, resulting in a global decrease in non-tagged proteins, or could also reflect the reduction in cell cycling that is typically observed as cells reached confluency. This is discussed in our revised manuscript (p 14, "In comparison, lipofectamine knock down levels remained low").

Reviewer 2

4. The approach of using a proteomics analysis to screen out the cytoplasmic proteins with strong binding affinities to PDMAEMA brush-grafted nanoparticles is a plus. However, the analysis did not provide any findings that were not known previously or unexpected, particularly in light of the qualitative nature of the conclusion "RNA displacement is clearly regulated by the strength and charge density of the macromolecules competing, as well as their molecular weight."

We disagree with this statement. Our data and model have enabled us to quantify the contributions of different molecules to competitive binding (Figure 3G). In addition, the impact of molecular competition on RNA release, but also the association of a range of cytosolic molecules that may have broader implications on the fate of delivery vectors, but also their impact on global translation, is directly evidenced by our proteomics analysis.

The types of protein sequestered at the surface of polymer brush vectors is particularly interesting and reveals an interactome dominated by RNA, RNA binding proteins (including ribosome associated proteins and translation factors), proteins associated with endosome and vesicular transport, as well as protein post-translational modification and proteolytic degradation. This not only points to the passive, yet tuneable release mechanism that is quantified in our study (competitive binding and differences observed between PDMAEMA, PMETAC and CS-PMETAC), but also to the potential impact that this proteome may underlie. Indeed, the impact that binding of ribosome complexes and translation/transcription factors by cationic nanoparticles (including factors associated with very broad range of functions such as YAP1 and YB1) may have on gene expression are unknown. It is possible that they may lead to off-target effects that could be detrimental to therapeutic strategies. Similarly, the association of proteins involved in stress response (e.g. heme-binding protein 2, E3 ubiquitin-protein ligase PPP1R11) and the proteasome may reflect some of the cytotoxic effects typically associated with polycationic vectors. Therefore, our work indicates that strategies that may allow to regulate association with these proteins (e.g. via limiting surface adsorption, whilst enabling RNA loading and delivery) could limit such unwanted processes. This is now discussed in our revised discussion section (p18-19, "The identification of the composition of the

polycationic..." and "Comparison of the cytosolic proteome associated with PDMAEMA and PMETAC ...").

5. Another limitation of this work is that the second half of the paper has minimal connection with the results from the detailed proteomics analysis. The concept of charge conversion carrier has been reported previously. The confirmation of the hypothesis on RNA release was only performed with RNA displacement test. What is the utility of the sophisticated analysis completed for the design or validation of the polycationic carriers?

The two parts of our manuscript are complementary. The design of brushes with strong binding affinities is particularly attractive to regulate molecular binding competition, owing to its simplicity, and affords high efficiencies of KD even after 10 days of culture. A quantitative model that enables to predict the impact of chemical modification and design on long term transfection efficiency will be of interest to the communities of biologists, bioengineers and clinical scientists interested in RNA delivery.

In addition, taking this comment into account, we have carried additional characterisation of the cationic vector proteome upon incubation in cytosolic fractions, in the case of both PDMAEMA and PMETAC brush-functionalised nanoparticles. This allowed us to identify significant similarities in the composition of associated proteomes, but also some differences (binding of histones, keratins and weakly acidic cytosolic proteins not associated with RNA binding or endosomes, the ER or the actin/microtubule cytoskeleton). This comparison therefore strengthens the potential generality of the proteome identified and its implications for gene delivery and release. This data is presented in our revised Figure 1 and Supplementary Figures S1, as well as Supplementary Tables S1 and S2. It is discussed in our results section its implications are presented in our revised discussion section (p18-19, "Comparison of the cytosolic proteome associated with PDMAEMA and PMETAC ...").

6. Establishing the molecular mechanism that regulates cytosolic dissociation of polycationic gene delivery vectors is valuable. What may be the limitation of using a whole cell lysate which may include all components inside cytosol, cellular vesicles and nucleus? The conclusion of this study seemed to suggest that RNA competition is the primary factor to consider for cell transfection process.

Many thanks for this suggestion. We agree that this was a limitation of our study. To address it, we have repeated our proteomics analysis, in the presence of cytosolic fractions. Cell lysate were fractionated directly after harvesting, by sequential centrifugation and ultracentrifugation. This afforded purified cytosolic extracts free of other sub-cellular components such as nuclei, fibrous cytoskeleton, mitochondria and vesicles. Following incubation with PDMAEMA and PMETAC brush functionalised nanoparticles, the resulting proteome was characterised again, via mass spectrometry. We identified comparable classes of proteins within these proteomes to those identified from full lysates, although with reduced number of proteins (perhaps as a result of the more dilute nature of the cytosolic fractions obtained). In addition to the differences observed between PDMAEMA and PMETAC cytosolic proteomes (see above), we also identify a number of key proteins involved in the assembly of ribosome complexes, RNA/DNA binding and translation/transcription. We propose that these not only reflect the nature of one important class of macromolecular competitors resulting in RNA desorption upon cytosolic entry, but potentially also point to a new origin for some of the off-target effects that are sometimes reported with polycationic vectors. This may therefore further guide the design of next generations of polymer brush delivery vectors. In addition to the additional data submitted

in our revised manuscript, we have expanded our discussion of the description of the proteome and its proposed implications.

7. The abstract does not capture the key findings of this paper and included only vague and qualitative statements.

We have revised our abstract to bring more quantitative conclusions of our study to the forefront. See our revised manuscript ("Our results identify the importance...").

8. The finding of adsorption of high IP proteins is interesting. Unfortunately, there was not follow up; the explanation on these proteins adsorbed "indirectly" to polycationic brushes is handwaving at best.

We took in the point made by the reviewer and proposed to address it in two different ways. First we confirmed again the presence of high IP proteins within the proteome, now directly associated with cytosolic extracts (revised Figure 1B). Second, we carried out secondary binding experiments in the presence of RNA molecules (Supplementary Figure S4). To this effect, we used tagged acidic (BSA) and basic (poly(L-lysine), PLL) and allowed these molecules to interact (separately) with RNA-loaded PDMAEMA-functionalised nanoparticles. We indeed observed significant recruitment of both types of proteins, therefore confirming that both low and high IP macromolecules can bind to RNA-loaded polymer brushes.

9. The statement on "polymer specific" adsorption is not sufficient. How would such a conclusion compromise the finding with a PDMAEMA brush-coated nanoparticles?

We understand that this comment refers to our conclusion regarding the regulation of the polycationic vector-cytoplasmic interactome (p 19, "Understanding the polycationic vector-cytoplasmic interactome..."). We fully agree that little specificity can be hoped from simple cationic polymer brushes, similarly to other cationic polymeric vectors (e.g. PEI, PLL, PDMAEMA, cationic dendrimers). However, such interactions can be modulated if hierarchical structures can be generated. For example block copolymer brushes such as PDMAEMA-b-POEGMA, which display excellent stability in serum, yet allowed RNA infiltration and capture, as we previously demonstrated. Such architecture, combined to suitable biofunctionalisation could be used to achieve more specific interactions with proteins or sub-cellular components. This has now been clarified in our revised conclusions (p 19, "For example, harnessing the ability (...) cytoplasmic interactions").

REVIEWERS' COMMENTS

Reviewer #1 (Remarks to the Author):

The revised manuscript entitled "Competitive binding and molecular crowding regulate the cytoplasmic interactome of non-viral polymeric gene delivery vectors" has been improved obviously. The additional experiments and related data have been added in the revised manuscript, and the several issues have been clarified according to the reviewers' opinions. The resubmitted manuscript is well structured and the points from reviewer's comments have been addressed in the manuscript. In my opinion, this paper could be accepted for publication.